# In vivo imaging with a water immersion objective affects brain temperature, blood flow and oxygenation

Morgane Roche[1], Emmanuelle Chaigneau[1], Ravi L Rungta[1], Davide Boido[1], Bruno Weber[2,3], Serge Charpak[1]*

[1]Laboratory of Neurophysiology and New Microscopy, INSERM U1128, Université Paris Descartes, Paris, France; [2]Institute of Pharmacology and Toxicology, University of Zurich, Zurich, Switzerland; [3]Neuroscience Center Zurich, University and ETH Zurich, Zurich, Switzerland

**Abstract** Previously, we reported the first oxygen partial pressure (Po2) measurements in the brain of awake mice, by performing two-photon phosphorescence lifetime microscopy at micrometer resolution (Lyons et al., 2016). However, this study disregarded that imaging through a cranial window lowers brain temperature, an effect capable of affecting cerebral blood flow, the properties of the oxygen sensors and thus Po2 measurements. Here, we show that in awake mice chronically implanted with a glass window over a craniotomy or a thinned-skull surface, the postsurgical decrease of brain temperature recovers within a few days. However, upon imaging with a water immersion objective at room temperature, brain temperature decreases by ~2–3°C, causing drops in resting capillary blood flow, capillary Po2, hemoglobin saturation, and tissue Po2. These adverse effects are corrected by heating the immersion objective or avoided by imaging through a dry air objective, thereby revealing the physiological values of brain oxygenation.
DOI: https://doi.org/10.7554/eLife.47324.001

*For correspondence:
serge.charpak@parisdescartes.fr

Competing interests: The authors declare that no competing interests exist.

## Introduction

The synthesis of new two-photon phosphorescent oxygen sensors such as PtP-C343 and the use of two-photon phosphorescence lifetime microscopy (2PLM) have revolutionized measurements of oxygen partial pressure (Po2) in the rodent brain (*Finikova et al., 2007*; *Finikova et al., 2008*; *Wilson et al., 2011*; *Esipova et al., 2019*). This approach has allowed mapping of Po2 in vessels and neuropil of the neocortex (*Sakadžić et al., 2010*; *Devor et al., 2011*; *Lecoq et al., 2011*; *Yaseen et al., 2011*; *Sakadžić et al., 2014*; *Lyons et al., 2016*; *Sakadžić et al., 2016*; *Kisler et al., 2018*; *Moeini et al., 2018*) and olfactory bulb (*Lecoq et al., 2011*; *Parpaleix et al., 2013*; *Lyons et al., 2016*), to a depth of several hundred micrometers. 2PLM's micrometer spatial resolution and the possibility to monitor simultaneously Po2 and individual red blood cells (RBCs) in capillaries revealed the presence of erythrocyte-associated transients (EATs), Po2 transients that are associated with individual erythrocytes (*Lecoq et al., 2011*). We reported that in anesthetized mice, Po2 between two EAT peaks is at near-equilibrium with Po2 in the neuropil surrounding capillaries, providing a unique and noninvasive way to measure brain oxygenation (*Parpaleix et al., 2013*). Using this approach in awake mice chronically implanted with a glass window and trained for several weeks in order to minimize stress, we then reported resting Po2 values of about ~25 mmHg in olfactory bulb glomeruli and in the superficial layers of the somatosensory cortex, that is from layer I to IV (*Lyons et al., 2016*). However, our chronic window preparation imposed removing the scalp and the cranial bone, which play an important role as thermal insulators and temperature regulators. As a result, the experiments were done under conditions of low brain temperature. The thermal

consequences of imaging through a glass window have been previously investigated in acute mice preparations, revealing a depolarization of neurons and changes of cortical network properties (*Kalmbach and Waters, 2012*; *Podgorski and Ranganathan, 2016*). These studies raise the questions of whether brain temperature is similarly lowered in animals chronically implanted with a glass window and of the extent to which this impacts blood flow dynamics and brain oxygenation.

Here, we investigated the relationship between temperature and blood flow parameters or oxygenation in small capillaries of the mouse olfactory bulb, anesthetized and awake. We show that i) resting brain surface temperature recovers to about 37°C within few days post-surgery and, ii) heat is constantly lost through the glass window chronically implanted over the olfactory bulb. 2PLM and standard two-photon imaging with a cool (room temperature) water immersion objective decreases brain temperature by ~2–3°C, causing a significant drop of resting capillary blood flow and red blood cell (RBC) velocity. It also lowers all Po2 parameters, measured after taking into account the temperature-dependence of the oxygen sensor PtP-C343. Heating the objective to correct brain temperature or imaging with a dry objective are required to determine the true physiological values of capillary blood flow and brain oxygenation. We further extended the study to mice with a reinforced thinned-skull window over the barrel cortex (*Drew et al., 2010*).

## Results

### Brain surface temperature under several imaging conditions

Mice were implanted with a chronic glass window, below which a small thermocouple (80 µm) was placed and fixed to continuously report temperature at the olfactory bulb surface. A second thermocouple was placed on the upper surface of the window during each imaging session (*Figure 1a1*). Mice were head-fixed, placed below the two-photon microscope and imaged through three types of objectives, a dry air 60X objective (NA = 0.7), a water immersion objective at room temperature (21–23°C, 60 X, NA = 1.1), and a similar water immersion objective heated with a temperature-controlled heating band (*Figure 1a2–a4* and *Figure 1—figure supplement 1*). Surgery associated with general anesthesia caused a temperature decrease at the brain surface that outlasted anesthesia and 2–4 days were necessary for brain temperature recovery to ~37°C (*Figure 1b*). All imaging experiments were thus performed after a minimum delay of 5 days. In awake animals, temperature immediately above the glass window was about ~35–36°C, indicating that the brain constantly loses heat to the room environment (maintained at 21–23°C). This energy loss was present in both awake and anesthetized mice, the temperature gradient across the window shifting to lower values during anesthesia (*Figure 1c*). Adding water between the window and the objective, caused a transient drop of the surface brain temperature by 5–10°C that partially recovered within few minutes. In awake mice, it stabilized at ~2°C below the resting value (without immersion) and temperature recovered to its resting value upon heating the objective (*Figure 1b,c*). During anesthesia, temperature followed the same changes during water immersion. Note that all measurements were done while the body core temperature was constantly maintained at ~37°C throughout the whole experiment.

### Temperature sensitivity of the 2plm oxygen sensor PtP-C343

As phosphorescence lifetime of PtP-C343 is temperature-dependent (*Finikova et al., 2008*), we first established PtP-C343 calibration curves at the mean temperatures observed in our various imaging conditions, prior to any in vivo imaging experiment. The 2P phosphorescence lifetime of PtP-C343 was measured in a custom designed calibration chamber, constantly stirred and at controlled temperatures (*Figure 2a*). Po2 and temperature were constantly monitored with external probes placed in the imaging chamber. The progressive exchange of air with nitrogen in the gas chamber separated from the imaging chamber by a gas permeable polytetrafluoroethylene membrane allowed to progressively lower Po2 from 159 to a few mmHg. Two-photon lifetime decays were collected simultaneously, and calibration curves were generated at four temperatures (32.4°, 34.2°, 35.7°, 37°C) that were then used in the following in vivo experiments. Note that calibration curves were shifted to the right when lowering temperature (*Figure 2b*), the insets indicating the necessity of using the adequate curve to convert the decay tau to Po2.

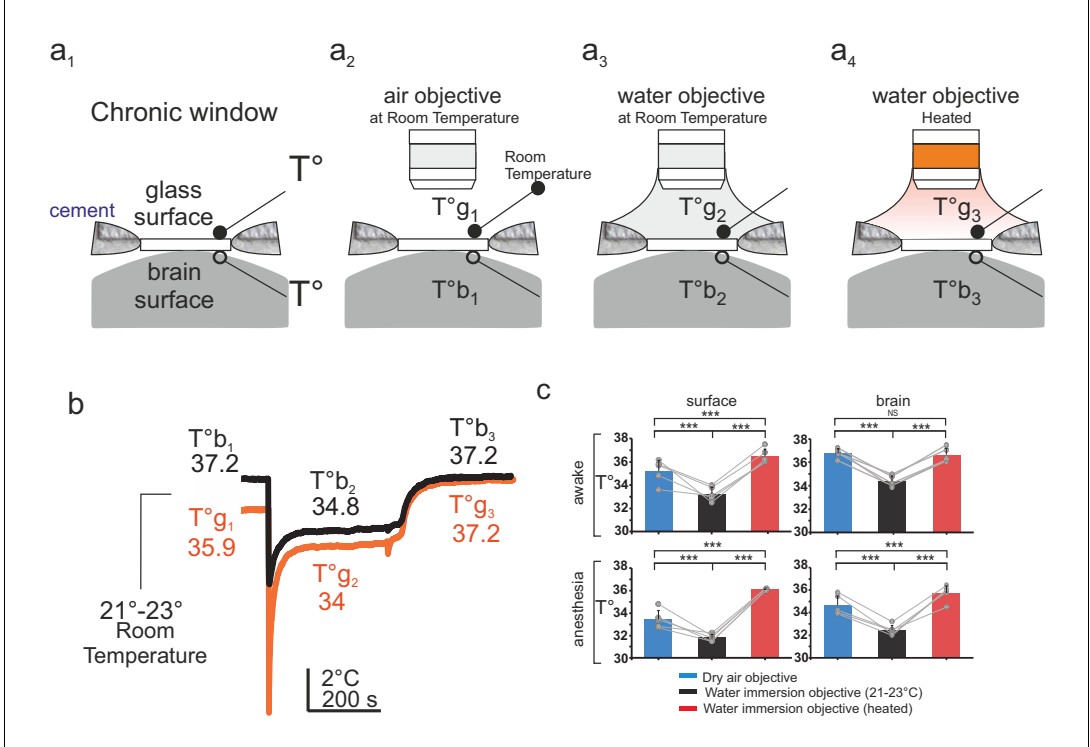

**Figure 1.** Brain temperature in different two-photon imaging conditions. (a1–a4), Schematics of 4 imaging configurations. (a1) Two thermosensors were positioned above and below the glass of the chronic window. 2P imaging was performed with (a2) an air 60x air objective (NA = 0.7) at room temperature (21–23°C), (a3) a 60x water immersion objective (NA = 1.1) at 21–23°C, (a4) a 60X heated water immersion objective. (b) In the absence of water, resting temperature is higher at the brain surface (T°b1) than at the glass surface (T°g1), indicating constant energy loss. The water immersion objective used at room temperature causes a transient drop of temperature at both sites (T°b2, T°g2) which stabilize with 5 min. Heating the objective restores brain and glass surface temperature to ~37°C (T°b3, T°g3). (c) 1-D scatter plots showing average glass and brain temperature values measured in all conditions in the same five mice (Data presented as mean ± s.e.m; Wilcoxon signed rank test; *** for p<0.001, NS: non significant). Note that in awake animals, brain temperature is physiological at rest and can be restored upon heating the water immersion objective.

DOI: https://doi.org/10.7554/eLife.47324.002

The following source data and figure supplement are available for figure 1:

**Source data 1.** Temperature with two objectives.
DOI: https://doi.org/10.7554/eLife.47324.004

**Figure supplement 1.** Objective heating system.
DOI: https://doi.org/10.7554/eLife.47324.003

## Brain temperature, resting cerebral blood flow, and oxygenation in different imaging conditions

Two-photon imaging was performed in small capillaries of the glomerular layer (diameter: 3.1+ /- 0.5, mean+/-std). Temperature changes due to two-photon scanning per se (*Podgorski and Ranganathan, 2016*; *Picot et al., 2018*) were detected by the brain surface thermocouple but not further considered, as they were minute in view of the changes causes by the water immersion objective. In awake mice imaged with the air dry objective, resting RBC velocity per capillary was 0.75 + /- 0.09 mm/s, resting RBC flow 44 + /- 5 RBC/s and resting Po2 Mean 42 + /- 3 mmHg. Imaging the same capillaries through the water immersion objective at room temperature decreased the three vascular parameters by ~33%, 22% and 20%, respectively (RBC velocity was 0.5 + /- 0.1 mm/s, RBC flow 35 + /- 5 RBC/s, Po2 Mean 34 + /- 3 mmHg), reaching values similar to what we previously reported under the same imaging condition (*Lyons et al., 2016*) (velocity n = 6 capillaries, three mice; blood flow and Po2 Mean n = 8 capillaries four mice). Interestingly, resting capillary hemodynamics and Po2 Mean recovered to their resting values upon restoring brain temperature to ~37°C (i.e. heating the objective and correcting for the working distance change, see Materials and methods) (*Figure 3a*). Analysis of EATs using our previous approach (*Lyons et al., 2016*; *Parpaleix et al.,*

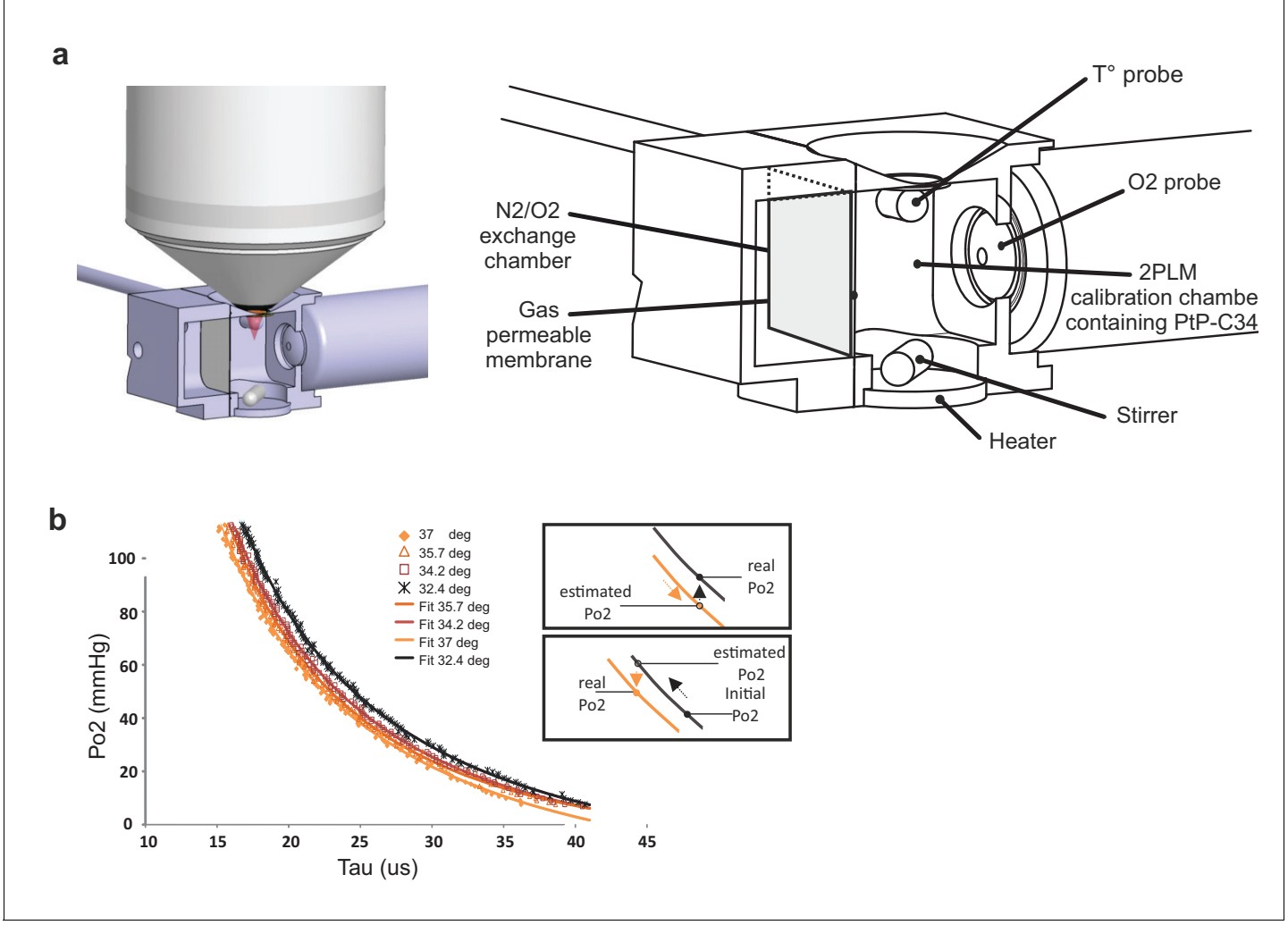

**Figure 2.** Calibration curves of PtP C343 at the temperatures measured below the glass window. (**a**) 3D design of the calibration chamber (left) with an enlarged diagram (right). The gas exchange chamber communicated with the 2PLM calibration chamber through a gas permeable membrane. The temperature of the 2PLM calibration chamber was continuously controlled and recorded. The Po2 sensor (PtP-C343) solution was constantly homogenized by stirring. An amperometric oxygen probe was used to monitor Po2. Two-photon phosphorescence lifetime decays and Po2 were simultaneously acquired. The exchange chamber was initially filled with air and temperature was set at either 32.4°C, 34.2°C, 35.7°C or 37°C. Then, N2 was gradually substituted to air, and Po2 decreased slowly in the calibration chamber down to few mmHg. (**b**) The calibration curve shifted to the right with lowering the temperature. The two insets show how 2PM Po2 measurements must be corrected when lowering (Top) or increasing (Bottom) temperature.
DOI: https://doi.org/10.7554/eLife.47324.005

*2013*), revealed that Po2 at the RBC border (Po2 RBC) and at mid distance between two RBCs (Po2 interRBC) (see *Figure 3b* inset) decreased by ~14% and~26% during imaging with a cool objective, an effect reversible upon heating the objective. As Po2 interRBC reports Po2 in the nearby neuropil (*Parpaleix et al., 2013*), these data demonstrate that resting tissue Po2 in glomeruli is about 31.4 mmHg (*Figure 3b*, middle). Given that Po2 RBC is representative of the Po2 level inside RBCs and using previous values of the Hill coefficient and P50 for mouse hemoglobin (*Uchida et al., 1998*), resting Po2 RBC values suggest that in olfactory bulb capillaries, resting hemoglobin saturation is about 70% and principally exists as oxyhemoglobin (*Figure 3b*, right).

Up to the reports *Lyons et al. (2016)* and *Moeini et al. (2018)*, 2PLM measurements of Po2 have been done in anesthetized rodents. Similarly, most in vivo studies using two-photon microscopy to investigate neurovascular coupling in the normal and pathological brain have also been performed under various types of anesthesia. We thus tested the reproducibility of our findings in mice

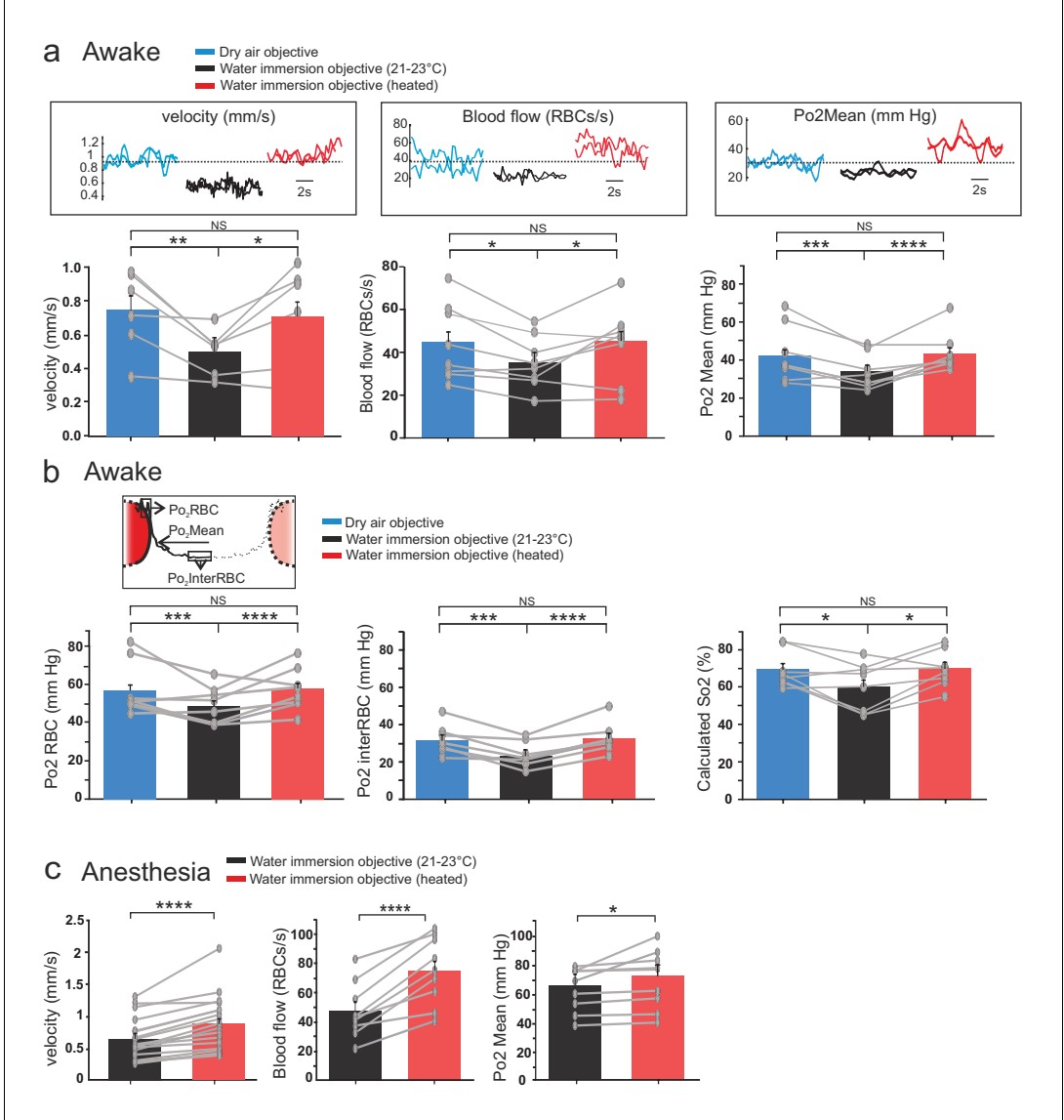

**Figure 3.** Effects of temperature on blood flow parameters and brain oxygenation in awake and anesthetized mice. (**a**) In awake mice, imaging with a cool water immersion objective decreases capillary RBC velocity (left, n = 6 capillaries, three mice), flow (middle, n = 8 capillaries, four mice) and Po2 Mean (right, n = 8 capillaries, four mice). The effect is reversible upon heating the objective. Upper insets illustrate typical velocity, flow and Po2 measurements. (**b**) Inset, Schematic of EATs: Po2 at the RBC border (Po2 RBC), Po2 at distance from a RBC (Po2 InterRBC) which gives an estimate of pericapillary Po2, average Po2 in the capillary (Po2 Mean). 1-D scatter plots show that lowering temperature reversibly decreases Po2 RBC, 'tissue' Po2 and RBC saturation. Note that all Po2 values are calculated using calibrations curves acquired at the corresponding temperatures. (**c**) In mice anesthetized with ketamine/medetomidine and breathing air supplemented with oxygen (30%), resting Po2 is high but still increases upon heating the objective, brain temperature reaching ~35.7 C° and Po2 ~70 mmHg (velocity, n = 18 capillaries, eight mice; blood flow and Po2, n = 9 capillaries, three mice). See Materials and methods for the statistical tests. *, **, ***, **** for p<0.05, 0.01, 0.001 and 0.0001 respectively, NS: non significant.
DOI: https://doi.org/10.7554/eLife.47324.006

The following source data is available for figure 3:

**Source data 1.** Blood flow parameters in awake mice.
DOI: https://doi.org/10.7554/eLife.47324.007
**Source data 2.** Blood flow parameters in anesthetized mice.
DOI: https://doi.org/10.7554/eLife.47324.008

implanted with a chronic glass window, anesthetized with a mixture of ketamine-medetomidine (100 mg kg−1 and 0.4 mg kg−one body mass, respectively) and supplemented with oxygen (reaching 30% in the inhaled air). Heating the objective had similar consequences: it significantly increased RBC velocity, RBC flow and Po2 in glomerular capillaries (*Figure 3c*). Note that with 30% oxygen inhaled (instead of 21%), resting Po2 Mean with the objective at room temperature (65 + /- 8 mm Hg) was much higher than that in awake mice breathing normal air (34 + /- 3 mm Hg).

## The thermal loss is not prevented by using a thinned-skull preparation

Thinned-skull cortical windows have been used for decades to image blood flow dynamics with laser Doppler flowmetry (*Gerrits et al., 1998*) or laser speckle (*Zakharov et al., 2009*). It has also become a very popular technique for imaging functional hyperemia using TPLSM, with the additional reinforcement of a glass cover slip (Polished and reinforced thinned skull (PoRTS)), which allows long term imaging of neurons and glial cells with improved imaging quality, stability and without inflammation (*Drew et al., 2010*). We had previously avoided the use of this approach for 2PLM measurements, as the oxygen sensor penetrates the thin layer of bone left below the glass and generates a two-photon surface excitation when imaging the tissue in depth. Nevertheless, as the bone layer could potentially diminish the temperature loss during imaging, we repeated experiments in awake mice with a glass reinforced thinned skull over the barrel cortex. The thermal sensor was placed below the thinned skull, in the cortex superficial layers (II to IV) (*Figure 4a*). As with the glass window preparation, 3–4 days were required for the brain to recover a stable brain temperature (37.7 + /- 0.2℃) (*Figure 4b*). Imaging with the water immersion objective at room temperature lowered the surface temperature by ~3℃ (34.5 + /- 0.4℃), an effect reversible upon heating the objective (36.9 + /- 0.4℃, n = 3 mice). Similarly, resting RBC velocity, RBC flow and mean Po2 increased by 18.4%,

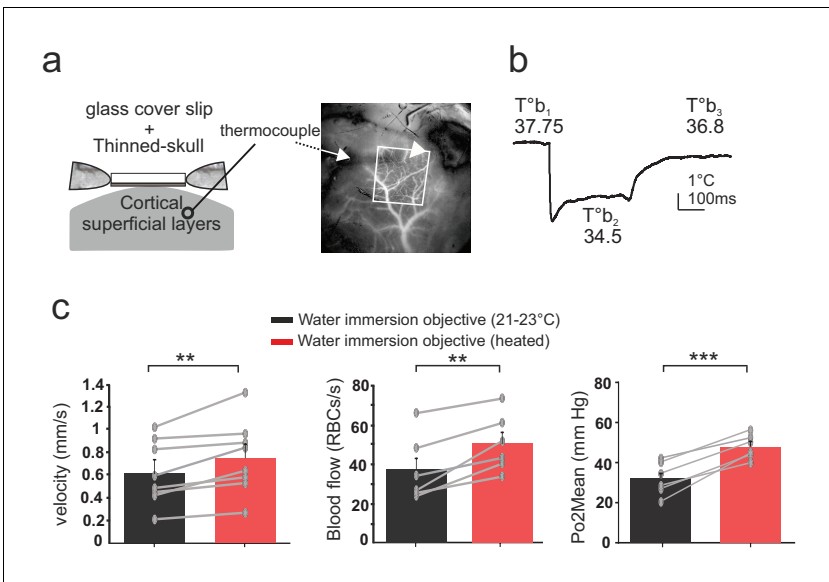

**Figure 4.** Effects of temperature in the barrel cortex of awake mice imaged through a reinforced thinned skull. (**a**) Left, schematics of the preparation. The thermosensor was placed in the cortex superficial layers (II to IV). Right, the sensor core and tips (white arrow and arrow head, respectively) are observable as a dark shadow below the bone and the surface vessels, that were injected with texas red and imaged through the reinforced thinned-skull with a stereoscope. (**b**) Imaging with the cool water immersion objective causes a decrease of brain temperature (T°b2), which partially recovers upon heating the objective (T°b3). (**c**) Resting Po2, blood flow and velocity increase upon heating the objective (velocity, n = 8 capillaries, four mice; blood flow and Po2, n = 6 capillaries, three mice). See Materials and methods for the statistical tests. **, *** for p<0.01, 0.001, respectively.
DOI: https://doi.org/10.7554/eLife.47324.009

The following source data is available for figure 4:

**Source data 1.** Thinned-skull.
DOI: https://doi.org/10.7554/eLife.47324.010

27.5% and 33%, respectively, upon temperature restauration (*Figure 4c*). We conclude that when imaging with the reinforced thinned-skull preparation, the thin bone layer does not protect against the thermal effect caused by the objective and immersion liquid.

## Discussion

The thermal effects of light during photoactivation (*Stujenske et al., 2015*; *Arias-Gil et al., 2016*; *Shin et al., 2016*; *Picot et al., 2018*) and two-photon imaging (*Podgorski and Ranganathan, 2016*) are widely recognized and there is a general consensus that light intensity should be kept as low as possible. In our study, we show that in order to provide true physiological values of brain oxygenation and resting vascular parameters, one has to correct a second adverse effect associated with two-photon imaging which is illumination independent but still involves temperature: the removal of the skin and part or all the skull which is required to image in the chronic window preparation. We find that as opposed to what occurs in the acute preparation, brain temperature recovers to 37°C after a few days, suggesting that, i) there is an adaptation of local blood flow, ii) the brain constantly loses heat to the air above the glass, as supported by the temperature gradient across the two sides of the glass window.

Adding water to the immersion objective tightly fixed to the microscope at room temperature, generates a large brain temperature drop that rapidly, but not fully, recovers and stabilizes to about 2–3°C below its physiological value. In both awake and anesthesia conditions, vascular and neuropil oxygenation decreased. This was not due to a temperature-dependent change of PtP-C343 triplet decay time as we adapted our calibration curves for each brain temperature. We believe that this drop in oxygenation principally results from the decrease of blood flow, rather than from minor changes of the affinity of hemoglobin to oxygen (*Dash and Bassingthwaighte, 2010*), the solubility (*Christmas and Bassingthwaighte, 2017*) and diffusion of oxygen. These findings reveal that our previous data acquired in the awake animal and in the same imaging conditions (*Lyons et al., 2016*) (with a water immersion objective at room temperature) have to be corrected and Po2 values increased. We now suggest that in glomeruli, the true physiological capillary Po2 Mean is about 42 mm Hg and Po2 Inter RBC, that is Po2 in the neuropil, about 31 mmHg. Similarly resting capillary blood flow and RBC velocity should be increased by 22% and 33%, respectively, considering imaging with a dry objective as the real control condition. Heating the objective is an alternative to imaging with an air objective. Furthermore, the quality of imaging with immersion objectives is superior and also require less laser power than with an air objective, an important consideration for avoiding heating, and potentially capillary stalls. Note that other approaches could have been used to warm the liquid: 1) perfusing warm saline on the window is an easy alternative (*Kalmbach and Waters, 2012*), although the suction noise could stress the mouse; 2) placing a heated plate directly on the skull (*Tran et al., 2012*) can raise the temperature below the objective to the right value, but with the risk of overheating the bone below the plate. As expected (*Zhu et al., 2009*; *Shirey et al., 2015*; *Suzuki et al., 2018*), anesthesia lowered resting brain temperature and cerebral blood flow. Imaging with the objective at room temperature further decreased the temperature to 32.4°C. As a result, heating the objective caused a larger temperature change than in the awake mouse, and had a larger effect on RBC flow, RBC velocity and Po2 Mean, even though the surface temperature was not corrected up to 37°C. This indicates that in all previous studies using TPLSM in anesthetized animals, including ours, resting vascular parameters (blood flow/velocity and Po2) below the glass window were not truly physiological even when global anesthesia was monitored carefully. Finally, although the reinforced thinned-skull window is a less invasive preparation with significant advantages for TPLSM imaging, unfortunately it does not ameliorate the thermal loss. In particular, as some phosphorescence can be generated at the level of the remaining bone, we do not advise the use of the reinforced thinned-skull for 2PLM measurements of Po2 in depth. To conclude, TPLSM and 2PLM modify brain activity, oxygenation and blood supply, unless temperature is corrected to its physiological value.

## Materials and methods

This section is brief, as the work is submitted as a Research Advance and has been done using exactly the same imaging apparatus, imaging analysis and animal preparations as in *Parpaleix et al. (2013)* and *Lyons et al. (2016)*. Only new materials and methods are described in detail below.

### Animal preparation and surgery

All animal care and experimentation were performed in accordance with the INSERM Animal Care and Use Committee guidelines (protocol numbers CEEA34.SC.122.12 and CEEA34.SC.123.12). Adult Thy1-GCaMP6f (GP5.11) mice from the Jackson laboratory) were used in this study (glass window: n = 4 mice for blood flow and Po2: three mice for RBC velocity in awake experiments, n = 3 mice for blood flow and Po2 and n = 8 mice for RBC velocity in anesthetized experiments; thinned-skull experiments: n = 4 mice for RBC velocity and n = 3 mice for Po2 and blood flow experiments), 3–12 months old, 25–35 g, both males and female, housed in 12 hr light-dark cycle). All experiments were performed in mice, anesthetized or awake, chronically implanted with a cranial glass window over the olfactory bulb. The animal preparation was done similarly to previously described (*Lyons et al., 2016*). For surgical anesthesia and anesthetized experiments, mice were anesthetized with a bolus of ketamine-medetomidine (100 mg kg−1 and 0.4 mg kg−one body mass, respectively) injected IP. Mice breathed a mixture of air and supplementary oxygen (the final inhaled proportion of oxygen was ~30%) and the body temperature was monitored with a rectal probe and maintained at ~37°C by a feedback-controlled heating pad. Mice were prepared with a reinforced thinned-skull over the barrel cortex as in *Drew et al. (2010)*.

### Habituation of mice to head restraint

Habituation sessions were performed similarly to previously described (*Lyons et al., 2016*) with few modifications. In brief, mice were supplied with a treadmill in the cage a week prior to surgery and restraint-habituation sessions started 3–4 days after surgery, the animal being head-fixed and free to run on the treadmill placed in the set-up that was later used for imaging. Habituation sessions were achieved 2–4 times per day over the course of one week, with the duration increasing from 5 min to 45 min. Each habituation session was carried out with scanning through the three objectives (a dry air 60X objective (Olympus, NA = 0.7), a water immersion objective at room temperature (21–23°C, 60X Olympus, NA = 1.1), and finally a similar objective heated with a temperature-controlled heating band (Okolab). The heating band did not cover entirely the objective and heating it to its maximum value (60°C) was necessary to warm up the objective and correct the brain surface temperature when the mouse was anesthetized. Animals were considered ready for use in experiments when they could be easily fixed in the recording apparatus while awake, and their behavior during the sessions consisted of short bouts of locomotion (~30 s) separated by periods of stillness (10 min) during which measurements were performed. Note that the training procedure was shorter than in *Lyons et al. (2016)* as we wanted to avoid any possible bouts of sleep.

### Brain surface temperature measurements

To measure temperature above and below the glass window, we used thermocouples with tips of about 80 μm (Omega 5TC-TT-KI-40–1M). One thermocouple was placed below the glass window (i.e. on the brain surface) and permanently fixed to the cranial bone during the chronic implantation of the glass window. In n = 6 animals, temperature was daily monitored with an amplifier for thermocouples (NI-9215, 16bits, 100 Hz sampling rate, National Instruments) driven by a custom software (Labview 2013, National Instruments), revealing that 2–4 days were required to recover a temperature of ~37°C. Temperature was measured in the three imaging conditions, with a dry objective, a water immersion objective at room temperature and a heated water immersion objective. The same protocol was used for the experiments in awake mice with a reinforced thinned skull window over the barrel cortex (n = 3 mice). In these animals, the thermocouple was placed in the cortex superficial layers (II to IV) and not at the surface.

### TPLM setup

The TPLM setup was custom-built and as described in *Lyons et al. (2016)*. In brief, femtosecond laser pulses were delivered by a Ti:Sapphire laser and targeted on the sample with galvanometric

mirrors, light intensity being controlled by an acoustic optical modulator. Fluoresceine dextran (Mw = 70 kDa) and the oxygen sensor, PtP C343 (Mw = 65 kDa) were excited at 890 nm and the emitted light was divided by a dichroic mirror (cut off wavelength = 560 nm) and collected on a green and a red sensitive PMT (Hamamatsu). PMT signals were amplified with custom-build electronics and sampled at 1.25 MHz by an acquisition card. Note that heating the objective changed the working distance by ~11 µm. Consequently, the focus was adjusted when vessels were imaged at two temperatures.

## Measurements of velocity RBC, blood flow and Po2

First, to measure blood flow and Po2 Mean simultaneously, the oxygen sensor, PtP C343 (100 µM in PBS) associated with Fluorescein dextran was injected intravenously via a retro-orbital injection. For awake experiments, animals were anesthetized a couple of minutes with isoflurane (4%, in air) and recovered for >1.5 hr before the experimental session began. Then we repeatedly excite a point in the capillary, once every 250 µs in capillaries. During the on phase of the AOM (25 µs), the RBC flow was extracted. During the off phase (225 µs), the phosphorescence lifetime of the oxygen sensor, the PtP-C343, was extracted, and Po2 was estimated from the phosphorescence decay. We discarded the first 5.6 µs after the end of the AOM gate. This applied to biological measurements as well as to the measurements done to establish the calibration curves. 40,000–80,000 decays were collected and averaged per point to determine Po2 Mean. Resting RBC velocity measurements were done using a line-scan recordings of 10–15 s. Po2 and RBC velocity analysis was performed using a custom written software (available with *Lyons et al., 2016*: https://github.com/charpak-lab/EAT-detection). During 2PLM experiments in mice with reinforced thinned-skull, we verified that in each case, some phosphorescence was generated in the thin bone layer. As a result, we limited our Po2 measurements to vessels from the upper part of layer 2/3, that is <180 µm from the surface.

## Erythrocytes-associated-transients (EAT) and saturation extraction

The method to extract EAT properties, RBC flow rates and Hemoglobin saturation (So2) can be found in *Lyons et al. (2016)*. The Po2 RBC value was calculated from the average lifetime of decays recorded in the 1–3.5 ms around the border of the RBC, whereas the Po2InterRBC was determined from average value of decays at mid-distance between RBCs (averaged over a window of at least 5 ms). So2 was estimated from Po2 RBC using the Hill equation, with a Hill coefficient (2.59) and P50 (40.2 mm Hg), which are accurate for C57BL/6 mice (*Uchida et al., 1998*) and have previously been used to make estimations of mouse cerebral So2 from Po2 data (*Sakadžić et al., 2014*; *Lyons et al., 2016*).

## Calibration of oxygen sensor

A dedicated two-compartment oxygen calibration chamber was designed and manufactured. The two compartments were tightly sealed and separated by a semipermeable polytetrafluoroethylene gas exchange membrane (Fluoropore 200 nm, Sigma Aldrich). The gas chamber had two holes, one for the gas inflow and a second for the outflow. Initially, the inflow contained 100% air and N2 was slowly introduced in order to cause a progressive change of Po2 in the second chamber, which was used to measure PtP-C343 phosphorescence quenching by oxygen. The imaging chamber had a volume of 1.08 ml, a liquid inflow and outflow to introduce PBS with the oxygen sensor, a heater, a stirrer, and tips of two control probes, one to measure O2 (OxyGold G Arc 120 - Hamilton) and the other temperature (PT100, class B, Roth Temperatursensorik). The top of the imaging chamber was sealed with a thin glass coverslip and placed below the microscope objective.

30000 decays were acquired for each Po2 Value. Calibration curves were established at four temperatures (32.4°, 34.2°, 35.7°, 37°C). The OxyGold electrode impulse response imposed to change Po2 in the imaging chamber very slowly, over several hours, in order to ensure that both 2PLM and chemical measurements report the same Po2.

## Statistical test

For all blood flow and oxygen data, statistical analysis was conducted using R (version 3.4.1, R Core Team, 2017). Linear mixed effects models (*Bates et al., 2015*) were used to analyze the differences between the Po2, saturation and RBC velocity values under different conditions. Each unique

combination of temperature (air objective, water immersion objective, heated water immersion objective) and position of measurement relative to RBC (in the case of Po2), was specified as a fixed effect, and intercepts for each mouse and capillary were specified as random effects. P-values for the overall significance of an effect were determined by likelihood ratio tests of the full model with the effect in question against a model without the effect in question. P-values for differences between groups were obtained post-hoc using the Tukey correction for multiple comparisons (*Hothorn et al., 2008*).

## Acknowledgements

Synthesis of the phosphorescent probe PtP-C343 was performed in the laboratory of Dr. Sergei Vinogradov (University of Pennsylvania) and supported by the grant R24 NS092986 'Enabling widespread use of high resolution imaging of oxygen in the brain' from the NIH USA'. We thank Stefan Weber and Harald Osswald for building/testing the calibration chamber, Kim Ferrari for help with statistics and Yannick Goulam for some temperature measurements. Support was provided by the Institut National de la Santé et de la Recherche Médicale (INSERM), the Leducq Foundation, and the ERC Advanced Grant 'Imaging-in-the-magnet'. The team of SC is part of the École des Neurosciences de Paris Ile-de-France network. BW acknowledges the financial support by the Swiss National Science Foundation.

## Additional information

### Funding

| Funder | Grant reference number | Author |
| --- | --- | --- |
| Fondation Leducq | | Morgane Roche |
| European Research Council | Imaging-in-the-Magnet | Serge Charpak |
| Swiss National Science Foundation | 310030_182703 | Bruno Weber |

The funders had no role in the decision to submit the work for publication.

### Author contributions

Morgane Roche, Data curation, Validation, Methodology, Writing—original draft, Conducted the in vivo experiments, Conducted the calibrations, Did the experiments using the thinned-skull preparation, Analyzed the data; Emmanuelle Chaigneau, Formal analysis, Methodology, Writing—review and editing, Conducted the calibrations, Analyzed the data; Ravi L Rungta, Methodology, Writing—review and editing, Did the experiments using the thinned-skull preparation; Davide Boido, Methodology, Analyzed the data; Bruno Weber, Software, Methodology, Writing—review and editing, Designed the calibration chamber; Serge Charpak, Data curation, Supervision, Funding acquisition, Validation, Investigation, Methodology, Writing—original draft, Supervised the study, Analysed the data

### Author ORCIDs

Emmanuelle Chaigneau https://orcid.org/0000-0003-4282-1774
Bruno Weber https://orcid.org/0000-0002-9089-0689
Serge Charpak https://orcid.org/0000-0002-5516-1245

### Ethics

Animal experimentation: This study was performed in strict accordance with the recommendations by Inserm. All animal care and experimentation were performed in accordance with the INSERM Animal Care and Use Committee guidelines (protocol numbers CEEA34.SC.122.12 and CEEA34.SC.123.12).

## Decision letter and Author response

Decision letter https://doi.org/10.7554/eLife.47324.014
Author response https://doi.org/10.7554/eLife.47324.015

## Additional files

### Supplementary files

• Transparent reporting form
DOI: https://doi.org/10.7554/eLife.47324.011

### Data availability

All data generated or analysed during this study are included in the manuscript and supporting files.

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
