## [Decision Letter]

Thank you for submitting your article "Two-photon imaging affects brain temperature, blood flow and oxygenation" for consideration by *eLife*. Your article has been reviewed by three peer reviewers, and the evaluation has been overseen by David Kleinfeld as Reviewing Editor and Timothy Behrens as the Senior Editor. The following individual involved in review of your submission has also agreed to reveal his identity: Andreas Linninger (Reviewer #3).

Charpak and colleagues show that the popular use of water immersion objections for in vivo imaging leads to a drop in the temperature of cortex by 2°C to 3°C. This occurs with imaging through either a closed cranial window or a thinned-skull transcranial window. Critically, this leads to a decrease in red blood cell flow and brain Po2. Presumably neuronal activity is similarly affected.

The reviewers have discussed the reviews with one another and the Reviewing Editor. We congratulate the authors on the care of their study. We are pleased to accept the manuscript subject to the following changes suggested by the reviewers and echoed by the Reviewing Editor.

1) Please specify the sample size and statistics for the data in the Table of Figure 1C.

2) It is important to know not just the mean changes in RBC speed and Po2 but the changes in distribution as well. Please replace the bar graphs in Figure 3A, B, C by Tukey box plot or 1-D scatter plot or even a Violin plot.

3) Please add details – perhaps even a supplementary figure, on the mechanics of heating the objective lens. Please comment on the ability to heat some of the very large (and thus massive) lenses in use these days

4) The title of the paper is misleading as the 2-photon imaging itself does not affect brain temperature. Please change it to reflect the fact that the immersion optic effects in vivo imaging, e.g., "in vivo immersion imaging affects brain temperature, blood flow and oxygenation".

The full reviews follow.

*Reviewer #1:*

This paper provides an important demonstration that 2-photon imaging through a cortical window with a water immersion lens lowers the brain temperature unless the objective is heated. Although heat loss through a cranial window has been previously demonstrated by Kalmbach and Waters, 2012, the present paper demonstrates that this has important consequences for measurements of CBF and cerebral oxygenation. It will be of wide interest to those studying brain blood flow.

In addition the authors are to be commended for extending their studies to include somatosensory cortex, which is studied more commonly than the olfactory bulb.

I recommend acceptance of this paper.

*Reviewer #2:*

The authors describe effects of cranial windows on brain temperature blood flow and oxygenation. The work is performed in mouse in vivo models and, importantly, compares anaesthetized and awake animals. These results add a cautionary tone to 2-photon microscopy experiments by showing that the preparation itself can significantly alter brain temperature when coupled with a water immersion objected and/or anesthesia. The authors show various combinations of imaging through cranial windows and thin skull preparations. In conclusion, any preparation that makes contact with a water immersion objective that will lead to cooling and alterations in blood flow. A surprising result is that a heated objective will not prevent the effects of hypothermia in the presence of anesthesia. The production of local cooling by an objective has been assumed for some time and previously reported by the Waters lab as cited by the authors.

The authors' results are consistent with previous reports in the literature. It is not surprising that connecting a small animal to a large objective through water immersion coupling will produce local temperature alteration. Nonetheless, the studies here do clearly define these temperature changes and how to potentially prevent them.

The authors' present brain temperature data in Figure 1 as mean +/- an error. There are no statistical test done here, so while it looks as though anesthesia conditions do not get back to 37 °C, there needs to be statistical testing to show that these are indeed different between awake and anesthesia conditions. The number of replicate animals should also be clearly stated and how the error was calculated (is this standard deviation overtime or between animals).

The authors should describe in some detail the system for heating the objective as this is an important step other investigators could take to rectify problems. Without this vital information, much of the utility of the manuscript is less certain. Although the authors describe that the heated objective is a solution, the question about animals under anesthesia is still difficult to reconcile (statistics here need to be done as it looks like a partial recovery). The authors may consider referencing previous work using 2-photon imaging and heated head plates to study stroke damage and recovery. Heating a stainless steel head plate using water perfusion was sufficient to maintain anaesthetized animals at 36.5 C, citation is Journal Cerebral Flow and Metabolism, 2012, vol.32, 437-442.

It is also conceivable that the inability to properly regulate temperature in anaesthetized animals could be a function of other factors, including the method used for head fixation and attachment, in particular the mass of this connection and how its conducting surface area contacts the animal.

*Reviewer #3:*

The authors present an experimental study evaluating the effect of temperature changes on oxygen measurements in two photon imaging. Based on measurements in a calibration chamber, they conclude that brain oxygen imaging through an open cranial windows or even thinned skull technique causes a 2-3 degree temperature decline that alters hemodynamic and metabolic parameters of the cortex. They clearly demonstrate the effects of temperature on in vivo measurements with and without temperature control as well as different skull preparation techniques.

The study offers two significant conclusions: (i) to avoid the effect of temperature changes on oxygen and blood flow, a water immersion objective should be used with the water bath kept at boy temperature of 37 °C. (ii) prior studies acquiring oxygen measurements without temperature control suffered from a temperature effect, which require correction.

The study is clearly written and the evidence is technically sound. I commend the author team for this study.

Technical comments:

My only question concerns RBC velocity (mm/s) and blood flow (RBC/s). With temperature changes, the authors observed is a slight increase in both quantities as measured in a few capillaries (n=6 to 8). How much does this increment compare to variations between different capillaries in the same animal and between similar caliber capillaries in different animals? Since there is variability of these quantities throughout the capillary network, it would be helpful to relate the temperature related trends to variability in the hemodynamic states of capillary vessels.

---

## [Author Response]

[…] We are pleased to accept the manuscript subject to the following changes suggested by the reviewers and echoed by the Reviewing Editor.1) Please specify the sample size and statistics for the data in the Table of Figure 1C.

We have removed the table and now present the temperature measurements for individual mice (the same 5 mice, in all conditions) in 1-D-scatter plots and the average of paired data as bar graphs.

2) It is important to know not just the mean changes in RBC speed and Po2 but the changes in distribution as well. Please replace the bar graphs in Figure 3A, B, C by Tukey box plot or 1-D scatter plot or even a Violin plot.

We thank the reviewers for pointing this missing information. We have added 1-D-scatter plots, with individual capillary values, to all bar graphs.

3) Please add details – perhaps even a supplementary figure, on the mechanics of heating the objective lens. Please comment on the ability to heat some of the very large (and thus massive) lenses in use these days

We have made a small supplementary figure with a schematics of the heating system. We have also made one measurement using a 25X objective (Olympus) borrowed from another laboratory. With this objective and the heating band that we have in the laboratory, we could correct the temperature drop, above the glass window, up to 37 °C, i.e. better than with the 60x objective. The reason is that the 25X objective is straight and better enwrapped by the heating band. This is indicated in the Results subsection “Brain surface temperature under several imaging conditions”.

4) The title of the paper is misleading as the 2-photon imaging itself does not affect brain temperature. Please change it to reflect the fact that the immersion optic effects in vivo imaging, e.g., "in vivo immersion imaging affects brain temperature, blood flow and oxygenation".

The new title is: “in vivo imaging with a water immersion objective affects brain temperature, blood flow and oxygenation.”

Reviewer #2:[…] The authors' present brain temperature data in Figure 1 as mean +/- an error. There are no statistical test done here, so while it looks as though anesthesia conditions do not get back to 37 °C, there needs to be statistical testing to show that these are indeed different between awake and anesthesia conditions. The number of replicate animals should also be clearly stated and how the error was calculated (is this standard deviation overtime or between animals).

We did not initially intend to statistically compare the temperature, at rest or during the drop, between anesthetized and awake animals. We understand that it was missing. As indicated now in the 1st response to the reviewing editor, we have substituted the table with a 1-D scatter plot containing the information. Note that under each condition, anesthesia or awake, there is a single measurement per animal (no replicate). To compare the data, we have now selected only the animals that were observed in all conditions (n=5). This new representation shows that resting temperature is indeed lower during anesthesia, as it has been previously reported by several groups.

The authors should describe in some detail the system for heating the objective as this is an important step other investigators could take to rectify problems. Without this vital information, much of the utility of the manuscript is less certain. Although the authors describe that the heated objective is a solution, the question about animals under anesthesia is still difficult to reconcile (statistics here need to be done as it looks like a partial recovery). The authors may consider referencing previous work using 2-photon imaging and heated head plates to study stroke damage and recovery. Heating a stainless steel head plate using water perfusion was sufficient to maintain anaesthetized animals at 36.5 C, citation is Journal Cerebral Flow and Metabolism, 2012, vol.32, 437-442.It is also conceivable that the inability to properly regulate temperature in anaesthetized animals could be a function of other factors, including the method used for head fixation and attachment, in particular the mass of this connection and how its conducting surface area contacts the animal.

We are describing and discussing in more detail the heating system (see also Figure 1—figure supplement 1). We are now citing the Murphy’s paper although we believe that the heated steel head plate changes brain temperature much more globally. There is a big risk that below the plate, temperature is above normal. We find important (as described in Figure 1—figure supplement 1) that the objective does not touch the cranium and that heat is transmitted to the window purely through the liquid (Discussion, last paragraph).

Reviewer #3:

[…] Technical comments:My only question concerns RBC velocity (mm/s) and blood flow (RBC/s). With temperature changes, the authors observed is a slight increase in both quantities as measured in a few capillaries (n=6 to 8). How much does this increment compare to variations between different capillaries in the same animal and between similar caliber capillaries in different animals? Since there is variability of these quantities throughout the capillary network, it would be helpful to relate the temperature related trends to variability in the hemodynamic states of capillary vessels.

We believe that reviewer #3 refers to Figure 4 (considering the number of capillaries indicated), where we measured the temperature effect in thinned-skull animals. In fact, the question concerns all experiments in awake animals. We have thus reanalyzed all of our recordings in awake animals (see Author response image 1). We find that the effect of temperature on RBC velocity (A, B) is distributed over all capillaries and mice. In addition, the average capillary diameter was 3.1 µm (+/- 0.5 µm, s.e.m.) in olfactory bulb. This corresponds to the small capillaries that are covered with long thin strand pericytes and do not actively dilate (see Rungta et al., 2018) (see Introduction, last paragraph; subsection "Brain temperature, resting cerebral blood flow, and oxygenation in different imaging conditions”, first paragraph. This “bias” results probably from the fact that these capillaries are more numerous. Moreover, imaging with a dry objective slightly lowers the image quality, it was more judicious to perform RBC velocity measurements in such vessels in which velocity is smaller and easier to measure. Note that the effect of temperature on RBC velocity is distributed among these capillaries (C, D) (see Author response image 1).
